# Tunable High-Q Factor Substrate for Selectively Enhanced Raman Scattering

**Zhengqing Qi [1],\*, Jinhuan Li [2], Peng Chen [1], Lingling Zhang [1] and Ke Ji [1]**

[1] School of Network and Communication Engineering, Jinling Institute of Technology,
Nanjing 211169, China
[2] Department of Basic Teaching, Nanjing Tech University Pujiang Institute, Nanjing 211134, China
* Correspondence: qizq@jit.edu.cn

**Abstract:** Most Surface-enhanced Raman scattering (SERS) substrates enhance all the Raman signals in a relative broad spectral range. The substrates enhance both the interested and background signals together. To improve the identification of target molecules from numerous background ones, substrates with multi high-quality (Q) factor resonance wavelengths can be designed to achieve the selective enhancement of specific Raman transitions. When the resonance frequencies are modulated to match the excitation and Raman scattering frequencies, the detection of the target molecule can be more effective. In this paper, we design a tunable high-Q SERS substrate with periodic silver bowtie nanoholes on silica spacer and silver film. The substrate possessed three high-Q and high electric field resonance modes, which resulted from the interaction of the localized surface plasmon resonance (LSPR) of the bowtie nanoholes, the surface plasmon polariton (SPP) of the period bowtie nanoholes and the Fabry–Perot (FP) resonance between the bowtie and silver film bottom. The interaction between these resonance modes resulted in not only a higher quality (Q) factor, but also a higher electric field, which can be employed to realize a potential substrate in high-sensitivity and selective-detection fields.

**Keywords:** SERS; high Q; selective Raman enhancement

## 1. Introduction

Raman spectroscopy has been widely utilized for its no marking, non-destructive testing and provides fingerprint information of the molecular structure. However, traditional Raman spectroscopy is extremely low, as the Raman scattering cross section of conventional molecules is $10^{-30}$ cm$^2$, which is only $10^{-6}$ of infrared processes and $10^{-14}$ of fluorescence processes [1,2]. To improve the Raman signals, surface-enhanced Raman scattering (SERS) is introduced to improve the signal intensity and has been applied to various fields, such as environmental pollution monitoring, food hygiene and safety, biomedicine, etc. [3,4]. The enhancement factor (EF) of SERS can be realized as high as $10^{14}$, which has great application prospects in high-sensitivity detection [5,6]. The enhancement mechanism is mainly contributed by the electromagnetic (EM) enhancement and chemical (CM) enhancement, while the EM enhancement plays a main role in the enhancement process [7,8].

Studies have found that the total EM enhancement in SERS is mainly determined by the product of the incident electric field intensity enhancement and the scattered field intensity enhancement [9,10]. Therefore, the SERS process can be divided into two stages: one is to enhance the excitation field of probe molecules and the other is to enhance the Raman scattering field of probe molecules. Then, the SERS enhancement is enhanced in terms of both the in-coupling and the out-coupling efficiencies. Most SERS systems exhibit a wide single resonance spectrum, achieving a high SERS enhancement when the broad

resonance spectrum overlaps the Raman excitation wavelength and the transitions of interest [6,11]. To obtain a higher SERS enhancement, structures with multi high-quality (Q) factor resonances can be designed to match different enhanced processes, respectively.

Various antennas have been introduced to realize multi wavelength resonance to realize selective Raman detection, such as the nanoring array, nanorice, nanorod dimer, nanohole array, etc. With these optical antennas, researchers have realized as high as $10^2$–$10^4$ SERS enhancement [12–14]. For example, Alam [15] selectively enhanced the R6G (Rhodamine 6G) with a double resonant nanodisk lattices. Shioi [16] also experimentally achieved the selective enhancement of R6G with a nanodisk array. In order to achieve a stable and highly selective Raman detection, it is necessary to consider the resonance region of different resonance wavelengths. When the field distributions of the excitation field and the emission field are basically coincided, the selected characteristic Raman signal of the target molecule can be significantly enhanced compared with other signals. Structures with significant "hotspots" that exhibit a similar energy distribution of different resonant modes can be introduced to realize a highly coincided field distribution.

Moreover, due to the inherently ohmic loss and electromagnetic radiation loss of metal, when the oscillating electrons on the metal surface interact with the optical wave, the excited plasmon resonance mode exhibits a relatively wide resonance bandwidth (50–100 nm) and the energy will attenuate rapidly [17,18]. These properties have adverse effects on the localization and enhancement of the optical field, and thereby affects the detection sensitivity and identifiability of the Raman signal. Accordingly, reducing the loss and improving the Q value of the optical antenna are effective methods to improve the recognition of target molecules. Due to the inherently lower Q of surface plasmon polaritons (SPPs), there are mainly two methods to improve the Q factor. One is to introduce a gain medium to adjust the refractive index of the surrounding environment around the optical antenna [19,20]. The other is to design a reasonable structure such as a cavity resonator, metamaterial or grating coupling to improve the Q factor significantly. This method can obtain a low loss, no loss and even plasmon radiation amplification antenna [21–23].

In this paper, we propose a high EF SERS substrate with three high-Q resonant wavelengths for selective Raman enhancement. The structure consists of period bowtie nanoholes upon silica spacer and silver film. By matching the first resonance mode to the exciting frequency and the other modes to the two selected specific Raman frequencies of the target molecules, the selective enhancement of target molecules can be realized while the overall enhancement of Raman signals is ensured. Furthermore, the resonant wavelengths can be modulated by optimizing the structure parameters. This selectively high-Q factor SERS substrate can successfully combine the high detection sensitivity and the recognizability of target molecules in the Raman detection system.

## 2. Simulation Models

The structure under investigation includes an array of silver bowtie nanoholes upon a silica spacer and silver film in a water environment. A schematic of the structure is shown in Figure 1a. The finite-difference time-domain (FDTD) method was used to calculate the spectrum response and field intensity of the structure. The normally incident light is polarized along the y direction. The surrounding material is set as water to reduce the dielectric permittivity difference, which results in a higher electric field localization and exhibits a potential substrate in the bio-detection field. The bowtie nanohole consists of the arm length L, taper angle $\alpha$, connected spacing d and thickness h1, as shown in Figure 1b. To satisfy the actual situation, the angles of the bowtie are smoothed with the fillet radius r = 10 nm. Additionally, the periodicity Px and Py, silica thickness h2 and silver thickness h3 are optimized by simulation. The silver film h3 is set to 150 nm to prevent LSPR coupling with the bottom side of the film. The dielectric permittivity of silver is taken from the experimental data by Johnson and Christy [24]. The refractive index of silica is set to a constant value of 1.45 with the neglect of frequency change.

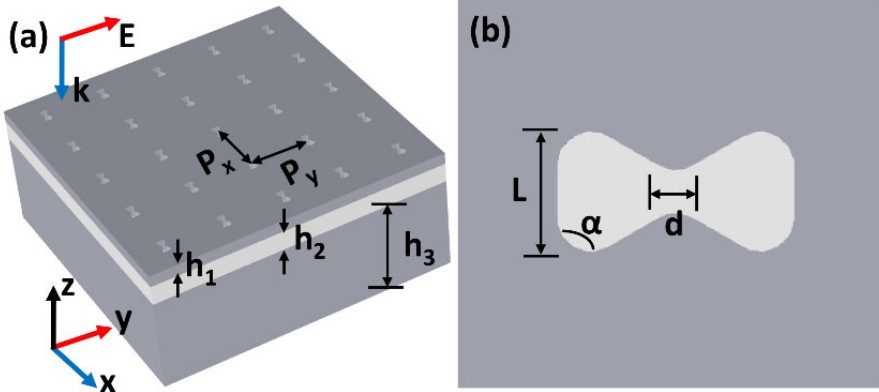

**Figure 1.** (**a**) Schematic of silver bowtie nanoholes structure. (**b**) Top view of the bowtie.

### 3. Simulation Results and Discussions

Before optimizing the structural parameters, three different silver nanostructures were placed upon the silica and silver film: a single disk, a single circular hole and a single bowtie nanohole. The far-field intensity distributions of the three structures were simulated to compare the scattering field properties. As shown in Figure 2a–c, it was found that all the three structures exhibited a good directivity due to the presence of the bottom silver mirror. In addition, the bowtie nanohole structure showed better properties in both field localization and directivity than the disk and circular hole cases. These properties provided a reliable precondition for the design of high-Q factor selective Raman enhancement substrates and effectively improved the detection sensitivity of the samples.

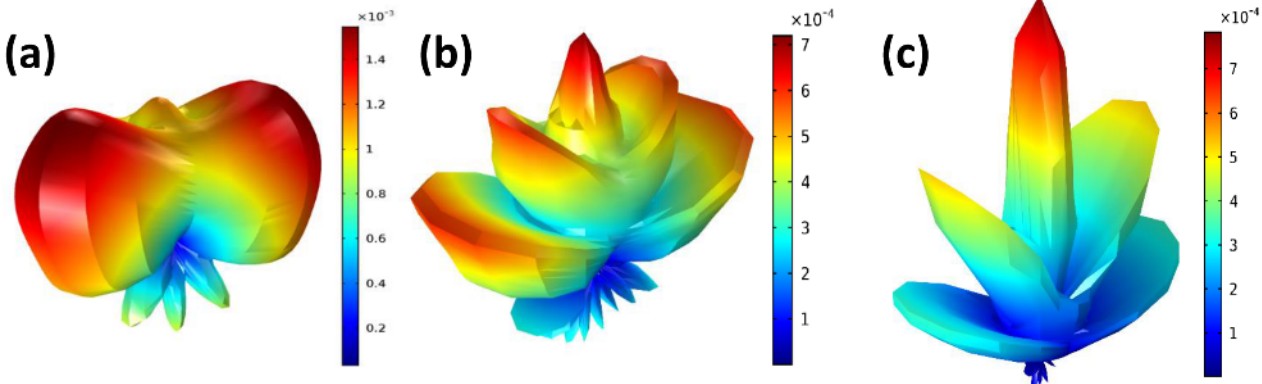

**Figure 2.** The far−field intensity distributions of a single (**a**) disk (radius r = 50 nm and thickness h = 30 nm), (**b**) circular hole (radius r = 50 nm and thickness h = 30 nm), (**c**) bowtie nanohole (arm length L = 65 nm, taper angle $\alpha$ = 50°, connected spacing d = 20 nm and thickness h1 = 30 nm) placed upon the silica (h2 = 30 nm) and silver film (h3 = 30 nm).

By setting the arm length L = 65 nm, taper angle $\alpha$ = 50°, connected spacing d = 20 nm and thickness h1 = 30 nm, the period constant Px = Py = 820 nm, silica thickness h2 = 45 nm, silver film h3 = 150 nm, we could obtain the electric field intensity that exhibited three high-Q resonant modes at the center of the bowtie gap, as the red square line shown in Figure 3. It can be seen that all the three resonant modes were enhanced by two orders of magnitude. By matching the first high-Q resonant wavelength to the exciting wavelength and the other two resonant wavelengths to the two selected Raman transitions, we were able to selectively enhance the selected Raman transitions by about $10^8$ simultaneously. Therefore, signals could only be identified from the background when the two selected

transitions were enhanced simultaneously. A traditional bowtie structure with a single wide resonance peak is introduced for comparison, with an arm length L = 60 nm, taper angle $\alpha$ = 60°, connected spacing d = 20 nm and thickness h1 = 30 nm, as the black circle line shown in Figure 3. It can be seen that the introduction of the period and bottom silver film formed a Fabry–Perot optical cavity and resulted in a strong constructive interference [25,26]. The FP coupling resulted in not only higher Q values, but also higher electric field intensities.

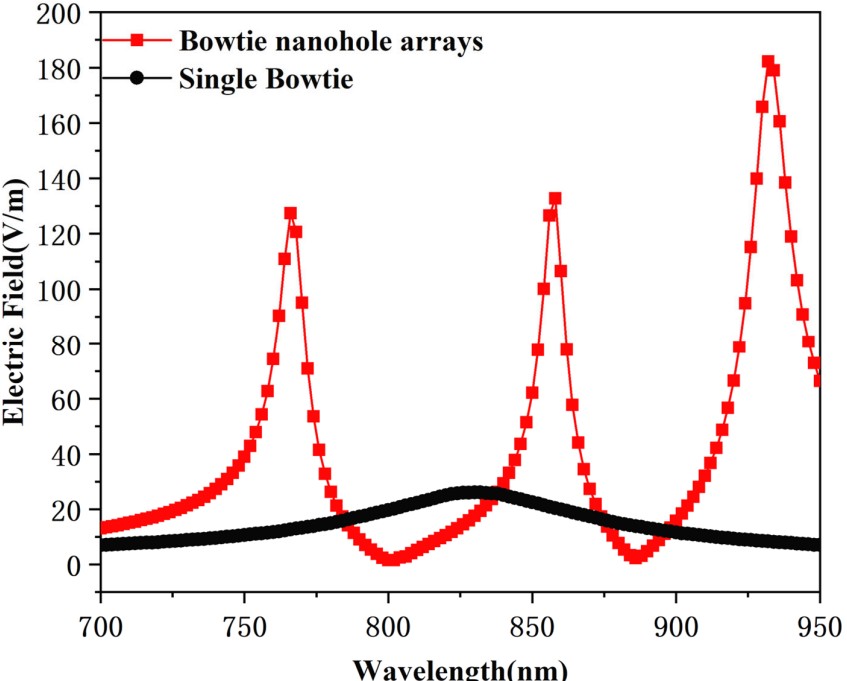

**Figure 3.** Simulated electric field at the center of the bowtie nanohole arrays (red square line) and the single bowtie (black circle line).

To analyze the near-field properties of the three resonance modes, the electric field distributions on the half-height surface are presented in Figure 4a–c. We can see that all the resonance modes exhibited a highly confined electric field distribution in the center of the bowtie gap, which provided a "hot spot" for signal detection. Compared with Figure 4a, both Figure 4b,c exhibit an obvious interference stripes distribution. Based on the field distribution characteristics, we could distinguish three resonant modes. The resonant mode at 766 nm (Figure 4a) was mainly confined in the bowtie gap, which corresponds to the LSPR mode; the resonant mode at 932 nm (Figure 4c) exhibited an interference fringe distribution, which corresponds to the SPR mode [8,25]; the resonant mode at 858 nm (Figure 4b) exhibited both localized and interference fringe distribution, which resulted from the FP coupling between the bowtie nanohole arrays and bottom silver mirror.

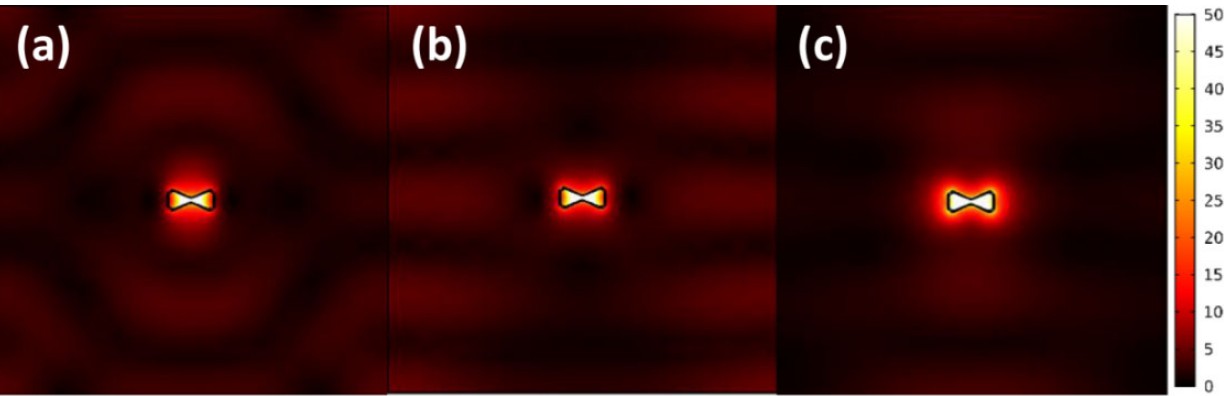

**Figure 4.** Intensity ($|E|^2$) distribution on the half-height surface of triple resonance substrate at (**a**) 766 nm, (**b**) 858 nm, (**c**) 932 nm

To further evaluate the quality of this substrate, the effective mode volume normalized to $(\lambda/2)^3$, $V_{eff}$, the Q value of the resonance mode and the ratio of these two values $Q/V_{eff}$ were calculated, as shown in Table 1. The $V_{eff}$ was calculated via an analogy with the effective mode lengths based on the theories in [27]. For the lossy metals, the electric field density at position $r$ can be expressed as $\mu_E = \varepsilon(\omega)|E|^2$, where $\varepsilon(\omega) = \varepsilon_1(\omega) + i\varepsilon_2(\omega)$ is the complex dielectric constant. Then, the effective mode volume can be expressed as:

$$V_{eff} = \frac{\int_V u_E(r)dV}{\max[u_E(r)]}$$

Here $\mu_{E(r)}$ represents the electromagnetic energy density at the position $r$. The Q value is calculated by estimating the full-width half maximum of the resonance with the fitting of Lorentzian functions [28]. It was found that the $V_{eff}$ and Q value at all the three resonance peaks of the structure exhibited the same order of magnitude, which ensured that the two selected Raman signals could be enhanced to almost the same extent and was convenient for the realization of the selective Raman enhancement. In addition, the traditional silver bowtie structure with a single resonance peak is introduced for comparison. It was found that due to the resonant coupling, the $V_{eff}$ of all the three resonant modes were effectively reduced and the Q values were also significantly improved by about one order of magnitude. Compared with the traditional bowtie structure, the bowtie nanoholes arrays significantly improved the Q value of plasmonic resonances, and its $Q/V_{eff}$ was increased by more than one orders of magnitude, providing a better SERS substrate for selective Raman enhancement.

**Table 1.** Normalized effective mode volume, the quality factor and $Q/V_{eff}$ for different resonance modes of bowtie nanoholes, and a single disk upon silica spacer and silver film.

| λ/nm | $V_{eff}$ | Q | $Q/V_{eff}$ |
|---|---|---|---|
| 766 | $1.55 \times 10^{-4}$ | 58.46 | $3.77 \times 10^5$ |
| 858 | $1.46 \times 10^{-4}$ | 70.83 | $4.85 \times 10^5$ |
| 932 | $0.94 \times 10^{-4}$ | 51.44 | $5.47 \times 10^5$ |
| Single bowtie | $3.34 \times 10^{-4}$ | 6.80 | $2.03 \times 10^4$ |

To realize the modulation for the detection of different target molecule, structural parameters were analyzed for the adjustment of the reflection spectrum. Here, we modulated the period, silica spacer and taper angle, respectively. As shown in Figure 5a, with the increase in the grating constant P, the resonant spectrum exhibited a relativistic red-

shift, which was caused by the decreasing coupling efficiency between the bowtie nanohole units. With the increase in the silica spacer h2, as shown in Figure 5b, the resonant spectrum exhibited a relativistic blueshift. Based on the mirror resonance theory, this was mainly caused by the increasing coupling efficiency between the SPP and its mirror resonance mode supported by the bottom silver mirror. With the increase in the taper angle $\alpha$, as shown in Figure 5c, the resonance spectrum also showed a relativistic blueshift, which indicated that the larger the structure is, the greater the blueshift exhibited. This phenomenon is in contrast with the traditional nanoparticles cases and guarantees the convenience of the bowtie structure preparation [9].

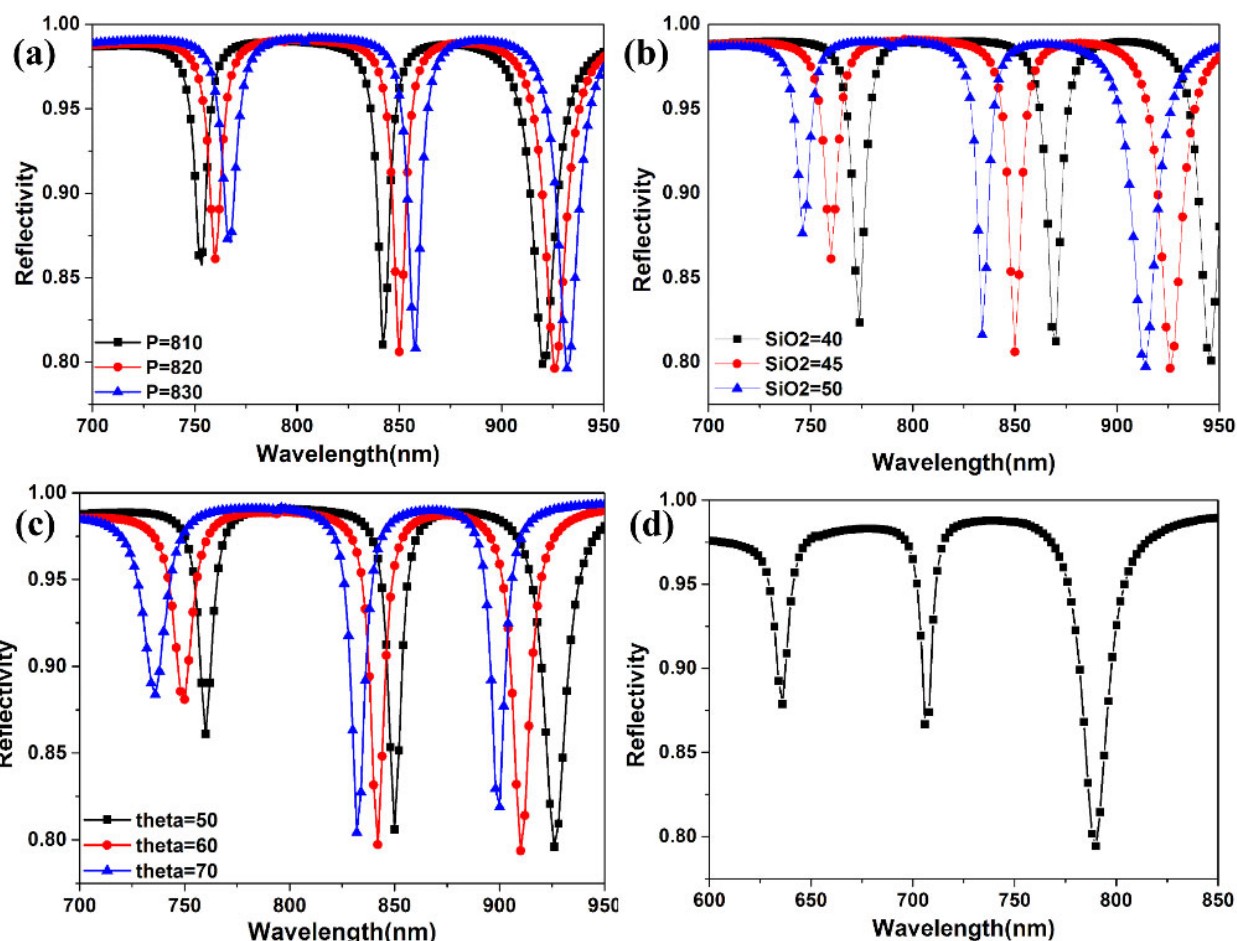

**Figure 5.** Simulated reflection spectrum of the bowtie nanoholes structure with the change in (**a**) period constant, (**b**) silica spacer thickness and (**c**) taper angle. (**d**) Bowtie nanoholes substrate in visible region with parameters L = 55 nm, $\alpha$ = 50°, d = 20 nm, h1 = 30 nm, Px = Py = 660 nm, h2 = 45 nm, h3 = 150 nm.

Considering all these structural parameters, the tunability of the substrate for different target molecules can be effectively realized, as shown in Figure 5d. By setting the arm length L = 55 nm, taper angle $\alpha$ = 50°, connected spacing d = 20 nm and thickness h1 = 30 nm, the period constant Px = Py = 660 nm, silica thickness h2 = 45 nm, silver film h3 = 150 nm, we could modulate the three resonance wavelengths to 630 nm, 705 nm and 790 nm, which were adjusted from the near-infrared to visible light and also exhibited a high-Q factor and high electric field enhancement. It could be utilized to selectively detect poisonous organic pollutants such as anisole, which exhibits the Raman transitions at 1600

$cm^{-1}$ and 3069 $cm^{-1}$. By setting the exciting laser wavelength to 630 nm and matching the two selected Raman transitions to 705 nm and 790 nm, respectively, the anisole signal could be selectively enhanced from numerous background signals.

Table 2 shows a brief survey of selective SERS enhancement substrates in previous works and our work. The gain factor is calculated based on the EM enhancement mechanism with the following expression: $G_{SERS} \propto |E_{exc}/E_{0exc}|^2|E_{sca}/E_{0sca}|^2$, where $E_{exc}$ and $E_{sca}$ are the local electric field of the exciting wavelength and scattering wavelength, respectively. We can see that the bowtie bole array exhibited better properties in both the gain factor and Q factor compared with the previous works, indicating a promising substrate for selective SERS enhancement.

**Table 2.** Comparison of selective SERS enhancement substrates in previous works and our work.

| Reference | Structure | $G_{SERS}$ | Q | Experimental Verification |
|-----------|-----------|------------|---|---------------------------|
| [13] 2010 | Nanodisk array and Gold film | $2.5 \times 10^6$ | ~30 | Yes |
| [12] 2011 | Mixed dimer array | $1.06 \times 10^5$ | ~10 | Yes |
| [16] 2014 | Nanodisk array and Gold film | $7.80 \times 10^7$ | ~20 | Yes |
| [25] 2019 | Nanohole array and Silver film | $4.36 \times 10^7$ | ~30 | Yes |
| [29] 2021 | Nanoring dimer array cavity | $6.71 \times 10^8$ | ~10 | No |
| Our structure | Bowtie hole array and Silver film | $5.39 \times 10^8$ | 70.8 | No |

### 4. Conclusions

In conclusion, we designed a tunable selective SERS enhancement substrate which owns three high-Q resonance modes. It is composed of periodic silver bowtie nanoholes on a silica spacer and silver film. The influence of different parameters, the period, silica spacer and taper angle, on the resonance frequencies are discussed. It was proved that with the adjustment of the different parameters, the resonance peaks can be flexibly adjusted and can maintain a strong electric field enhancement and high-Q factor. This tunable high-Q triple-resonance-modes substrate exhibits three advantages. Firstly, the total Raman signals can be enhanced about $10^4$ times for the enhancement of the exciting field. Secondly, the selected Raman frequencies matched with the resonance modes can be selectively enhanced about $10^4$ times compared with the mismatched ones. Considering the exciting process, the Raman signal is improved about $10^8$ times. Finally, considering the two selected transitions together, the identification of the target molecule can be further improved. Due to the properties of the substrate, i.e., tunability, high-Q and strong electric field enhancement, it provides a potential platform for Raman studies of various molecules.

**Author Contributions:** Conceptualization, Z.Q.; methodology, Z.Q.; software, Z.Q.; validation, K.J.; investigation, J.L.; writing—original draft preparation, Z.Q.; writing—review and editing, Z.Q.; supervision; funding acquisition, Z.Q., P.C. and L.Z. All authors have read and agreed to the published version of the manuscript.

**Funding:** This work was supported by the Ph.D. Supporting Project of the Jinling Institute of Technology (Grant No. jit-b-202101), the Hatch Project of the Jinling Institute of Technology (Grant No. jit-fhxm-2001), the Natural Science Foundation of Jiangsu Province (Grant No. BK20190111) and the Natural Science Research Project of Higher Education Institutions of Jiangsu Province (Grant No. 20KJB510013).

**Institutional Review Board Statement:** Not applicable.

**Informed Consent Statement:** Not applicable.

**Data Availability Statement:** The data that support the findings of this study are available from the corresponding author upon reasonable request.

**Conflicts of Interest:** The authors declare no conflict of interest.

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
