# Peer review of "Tunable High-Q Factor Substrate for Selectively Enhanced Raman Scattering"

_photonics, doi:10.3390/photonics9100755_

Round 1

Reviewer 1 Report

This work reports on the design of high quality (Q) SERS substrate for selective Raman scattering based on FDTD simulations. The authors tried to optimized the structure of periodic silver bowtie nanohole arrays on silica spacer and silver film, so that the LSPR and SPP effect can be readily adjusted and strong electric field enhancement can be achieved. I think the idea is interesting in view of proposing a high Q SERS substrate. However, this work is not suitable for “Photonics”. The authors can improve their work from the following aspects.

1. The language, style and grammar need to be improved. I see lots of serious mistakes both in the abstract and in the text that hindered my reading process.

2. The definition of “Q” is not clear. The authors quantified Q by estimating the full width half-maximum of the resonance with the fitting of Lorentzian functions. I don’t quite understand the scientificity. What is the definition of Veff?

3. What is the optimal structure herein?

4. What is the selectivity? I see limited connection between the FDTD simulations and specific Raman frequency of target molecules, and selective enhancement of target molecules, as mentioned in the introduction.

Reviewer 2 Report

The paper by Zhengqing Qia et.al. “Tunable High Q Factor Substrate For Selectively Enhanced Ra-man Scattering” describes a computational design of a SERS substrate with improved Q factor in three specific regions of excitation. The paper utilizes the periodic silver bow tie nanohole design with underlying silica spacer and silver film. Based on the reported results, the Q factor of such a substrate is dramatically improved for the following three wavelength of excitation: 766 nm, 858 nm, and 932 nm. Moreover, changing the parameters of nanohole shape and the separation it is possible to tune these wavelengths to the desired wavelength. The paper needs English revision as there is a number of misprints and errors. One thing that is missing and the paper would definitely benefit from is an example of what might be those molecular compounds which would benefit from the proposed design. At the very least the authors might consider taking an example from already published paper. 

Reviewer 3 Report

In this work Qi et al. proposed substrates for SERS measurements, that will allow for the amplification of Raman bands having specific frequencies. In my opinion, it is a very original idea - usually, to obtain selective SERS substrates, a different solution is used - the surface of the SERS substrate is modified in such a way that only some compounds can adsorb on the substrate. I am not able to fully check the correctness of this work, because I am not a theoretician, and this work is theoretical. But it seems to me that everything is fine with this work. Although it can be easily calculated from the presented data, I think that it would be very useful to provide the frequencies of the SERS bands that will be amplified in a selective way using the SERS substrates proposed.

Author Response

Thanks for your comments. Due to the limit of laboratory conditions, we conducted the theoretical studies only. Nest step, we will try to conduct the experimental validation with cooperation units.

Round 2

Reviewer 1 Report

I see the authors tried to address the issues. However, the issues remain unsolved, including the syntex, the undefended conclusions. The authors need to demonstrate their proposals by conducting some experiments with their collaborators. I don't think this work can be published in its current form.

Round 3

Reviewer 1 Report

This work has been improved, it can be published in the journal of Photonics.